# Cord blood IgA/M reveals in utero response to SARS-CoV-2 with fluctuations in relation to circulating variants

Olivier Pernet [1] ✉, Toinette Frederick[1], Amila Adili[2], Jay Hudgins[3], Patricia Anthony[1], Gwyndolyn McCaney[1], Wendy J. Mack [2,4], Eunice Noriega[1], Jennifer Lopez[1], Steven Balog[1], Manoj Biniwale[5], Amy Yeh[5], Allison Bearden[1], Rangasamy Ramanathan[5] & Andrea Kovacs [1,3,4] ✉

It is estimated that in utero SARS-CoV-2 infection is rare. However, few studies have systematically assessed for IgA and IgM antibodies indicating potential in utero response to SARS-CoV-2 infection using multi-isotype serology, and no studies have assessed in utero infection markers in relation to circulating variants. Between October 21, 2021 and February 15, 2023, remnant cord blood samples (CBS) from neonates born at a single hospital in Los Angeles, were systematically tested for serological markers suggesting in utero infection. SARS-CoV-2 specific fetal IgA and/or IgM antibodies were detected in 28.7% (298/1038 CBS, 95% CI: 26.0, 31.6), higher than previous in utero infection estimates that used only PCR and/or IgM. Importantly, the probability of detecting markers of in utero infection varied by month (P-value = 0.0144). The prevalence of fetal IgA/IgM varied with the emergence of new variants, increasing during the BA.1 wave with a peak in February 2022 at 36% (18/50, 95% CI: 22.7-49.3) and again during the BA.4/5 wave, with a peak at 48.8% in September 2022 (39/80, 95% CI 37.8-59.7), suggesting variant-related fluctuations. These data suggest it may be useful to identify SARS-Cov-2 in utero exposure at birth so these newborns may be more closely followed for adverse clinical outcomes.

SARS-CoV-2 infection has caused millions of deaths and significant morbidity worldwide. Pregnant women and their newborns have been particularly vulnerable[1,2]. Recent studies suggest that some newborns exposed in utero to SARS-CoV-2 have developmental abnormalities[3,4] furthering the need to identify SARS-Cov-2 in utero exposure at birth so they may be more closely followed. Few studies have comprehensively tested for both IgM and IgA antibodies[5–7] to identify newborns

with possible in utero SARS-CoV-2 infection. The World Health Organization's (WHO) definition of confirmed in utero infection requires evidence of maternal SARS-CoV-2 infection during pregnancy with in utero fetal exposure, and SARS-CoV-2 persistence or immune response in the neonate[8] There are, however, significant challenges in diagnosing in utero infection[9,10]. PCR at birth may miss resolved infections, since maternal SARS-CoV-2 infection can occur any time prior to and

[1]Maternal, Child, and Adolescent Center for Infectious Diseases and Virology, Division of Pediatric Infectious Diseases, University of Southern California, Keck School of Medicine of USC, and Los Angeles General Medical Center, Los Angeles, CA, USA. [2]Southern California Clinical and Translational Science Institute (SC-CTSI), University of Southern California, Los Angeles, CA, USA. [3]Department of Pathology and Laboratory Medicine, University of Southern California, Keck School of Medicine of USC, and Los Angeles General Medical Center, Los Angeles, CA, USA. [4]Department of Population and Public Health Sciences, University of Southern California, Keck School of Medicine of USC, Los Angeles, CA, USA. [5]Division of Neonatal Medicine, University of Southern California, Keck School of Medicine of USC, and Los Angeles General Medical Center, Los Angeles, CA, USA. ✉e-mail: pernet@usc.edu; akovacs@usc.edu

throughout pregnancy, but viral replication is limited to a few weeks. Furthermore, because of their transitory nature, fetal IgM antibodies may only be present for a few weeks and become undetectable at delivery. SARS-CoV-2 specific IgA found in newborn cord blood may represent in utero infection occurring earlier in pregnancy, but no large study has evaluated newborns for both IgA and IgM. Evaluating for IgA may be advantageous[11,12].

Anti-Receptor Binding Domain (anti-RBD) antibodies are the major neutralizing antibodies that develop after vaccination and/or natural infection[12,13]. In the U.S., FDA approved vaccines all provide Spike glycoprotein immunogenicity, which includes anti-RBD, while anti-N (Nucleocapsid) antibodies are only detected with past infection.

In this study, we tested 1405 available remnant cord blood samples (CBS) of the 1561 newborns delivered at Los Angeles General Medical Center (LA General) between October 2021 and February 2023 for SARS-CoV-2 specific anti-RBD and anti-N IgG, IgA, and IgM. Because studies show most pregnant women are asymptomatic or have mild symptoms[14], we screened all newborn cord blood samples for IgG anti-RBD and anti-N indicative of maternal past infection and/or maternal vaccination (anti-RBD positive only)[15–18] using a multiplexed semi-quantitative assay[13,19]. For CBS with confirmed maternal past infection, defined as both IgG anti-RBD and anti-N above the positivity threshold of 700 Median Fluorescence Intensity (MFI), as defined by the manufacturer guidelines, we further tested CBS for SARS-CoV-2 specific IgA and IgM antibodies targeting RBD and N antigens[19,20]. Because antibody levels decline over time, and can vary with gestational age and vaccination status[21,22], we also tested all CBS with IgG anti-N above background level.

Our goal was to determine the prevalence of fetal serological response to SARS-CoV-2 suggesting in utero infection and assess if there are differences in these rates depending on circulating variants. We hypothesized that our multi-isotype assay that includes both IgA and IgM would identify newborns with evidence of fetal immune response and potential in utero infection, including infants who would otherwise not be identified using only IgM. We also hypothesized that we would see variation in the detection of positive IgM or IgA in cord blood specimens as new SARS-CoV-2 variants evolved, such as delta and omicron and its sub-lineages. As new highly infectious variants may emerge in the future, despite vaccination and/or past infection, it would be advantageous to have a screening tool to identify newborns with possible in utero infection.

## Results

### SARS-CoV-2 specific IgG
In total, we screened 1405 newborn CBS for SARS-CoV-2-specific IgG anti-RBD and anti-N (Fig. 1a), representative of maternal IgG profile any time prior to delivery, including prior to pregnancy[15–18]. Among the tested samples, 1269/1405 (90.3%) were positive (≥700 MFI) for anti-RBD, and of these 798/1405 (56.8%) had both anti-RBD and anti-N indicating past SARS-CoV-2 infection. Additionally, 287/1405 (20.4%) were positive for anti-RBD, but had anti-N below background threshold (<300 MFI), suggesting these samples came from newborns with vaccinated mothers with no serological evidence of past infection.

During our study period, Delta and Omicron variants emerged and spread through the Los Angeles County population. The prevalence of maternal infection (both IgG anti-RBD and anti-N ≥ 700 MFI) increased as the pandemic progressed from 22.5% (36/160) during the first 45 days of the study to 80.3% in the last 45 days (135/168) (Fig. 1b).

### SARS-CoV-2 specific IgM and IgA in cord blood
Altogether, a total of 1038/1405 (73.9%) CBS had IgG anti-N levels above background (≥300 MFI) (Table 1). These samples were further tested for IgA and IgM to evaluate for potential in utero infection (Fig. 1c). We first established and validated positivity thresholds for IgA and IgM anti-N and anti-RBD using 103 cord blood samples collected

before the emergence of SARS-CoV-2 (Supplementary Fig. S1), as described in Methods. We then tested the 1038 CBS: all yielded results for IgA, and 1035 for IgM (Table 1, Supplementary Fig. S2). In utero response to SARS-CoV-2 as defined by the presence of anti-SARS-CoV-2 specific IgA, or IgM, or both, was detected in 298/1038 newborns (28.7%, 95% CI: 26.0, 31.6) (Table 1, Fig. 1C). Of these 298 samples, 31 (10.4%) were positive only for IgM antibodies, 224 (75.2%) had only IgA, 40 (13.4%) had both IgM and IgA, and 3/298 (1%) were positive for IgA with unknown IgM (Table 1).

### Relationship between levels of maternal IgG anti-RBD and anti-N, and neonatal IgM and IgA
We further explored the relationship between maternal IgG anti-RBD and anti-N levels and detectable IgM and/or IgA. CBS with only IgA antibodies had higher median IgG anti-RBD levels (15,733 MFI) than those with only IgM antibodies (8140 MFI) (Z-statistic = −4.7209, $P = 0.0003$) and those with both IgM and IgA antibodies (12,620.5 MFI) (Z-statistic = −2.5257, $P = 0.0363$) (Fig. 2, Supplementary Table S2). Median IgG anti-N levels were also significantly greater for those with IgA only compared to those with IgM only (Z-statistic = −3.4971, $P = 0.0018$) (Fig. 2, Supplementary Table S3).

### Rates of fetal serological response to SARS-CoV-2 over the study period
Rates of fetal serological response to SARS-CoV-2 as evidenced by positive IgA and IgM in CBS, varied during the study period (Table 2 and Fig. 3). Overall, the probability of detecting markers of in utero infection varied by month. (Chi-Square = 30.78, DF = 16, $P$-value = 0.0144). At the end of the Delta wave, the prevalence of positive IgA/IgM in cord blood was 22.2% (4/18; 95% CI: 3.0–41.4) in October 2021 and 16.2% (6/37; 95% CI: 4.3–28.1) in November 2021. With the rise of Omicron sub-lineages, notable peaks occurred. Prevalence rates first increased from February to April 2022, following the BA.1 sub-lineage emergence, with a peak at 36% (18/50; 95% CI: 22.7–49.3). The rate decreased during BA.2 to 24.7% (18/73; 95% CI: 14.8–34.5) in May 2022 and 22.8% (13/57; 95% CI: 11.9–33.7) in June 2022. An increase was again observed during the BA.4/5 wave, with a peak at 48.8% in September 2022 (39/80; 95% CI: 37.8–59.7), and again in January 2023 at 37.0% (40/108; 95% CI: 27.9–46.1), during the transition from BQ.1 (and related sub-lineages) to XBB.1.5 (Fig. 3 and Table 2).

## Discussion
Previous studies suggest that in utero SARS-CoV-2 transmission occurs infrequently[1,5]. Nevertheless, some infants exposed in utero are reported to have developmental abnormalities and other morbidities[1–4]. Unrecognized in utero infection may explain this discrepancy. To date, no published studies have assessed CBS for IgM and IgA SARS-CoV-2 specific antibodies as markers for possible in utero infection in a large cohort. Further, no study has examined if there are variations in rates of potential in utero infection as new SARS-CoV-2 variants evolve.

To date, our study is the largest study to assess for serological markers suggesting in utero infection at a population level in a single hospital where 90% of newborn CBS were tested for SARS-CoV-2 specific anti-RBD and anti-N IgA and IgM antibodies. There are several notable findings. First, we found a higher rate of fetal response to SARS-CoV-2 (28.7%) using SARS-CoV-2 specific IgA and/or IgM antibodies in cord blood than previous estimates of in utero infection based on PCR and IgM antibodies. Importantly, only 6.8% (71/1038) of newborns would have been identified if only tested for IgM. Second, we found that IgG anti-RBD levels were significantly associated with the timing of infection: newborns with SARS-CoV-2 specific IgM, indicative of more recent infection, had significantly lower IgG levels compared to newborns with IgA only. Altogether, our results suggest that IgM testing alone is insufficient to identify potential in utero infection and testing should include both IgA and IgM.

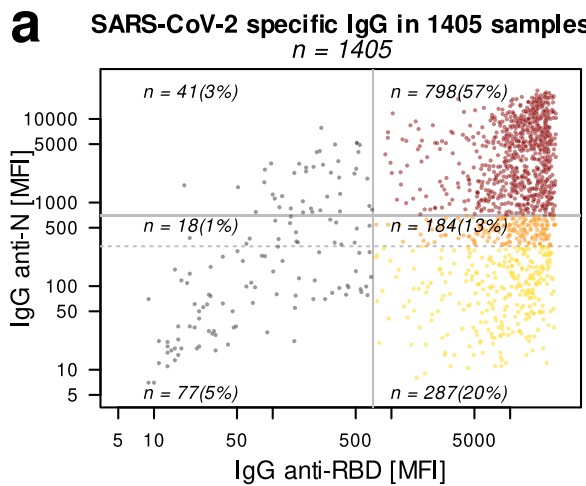

**a   SARS-CoV-2 specific IgG in 1405 samples**
n = 1405

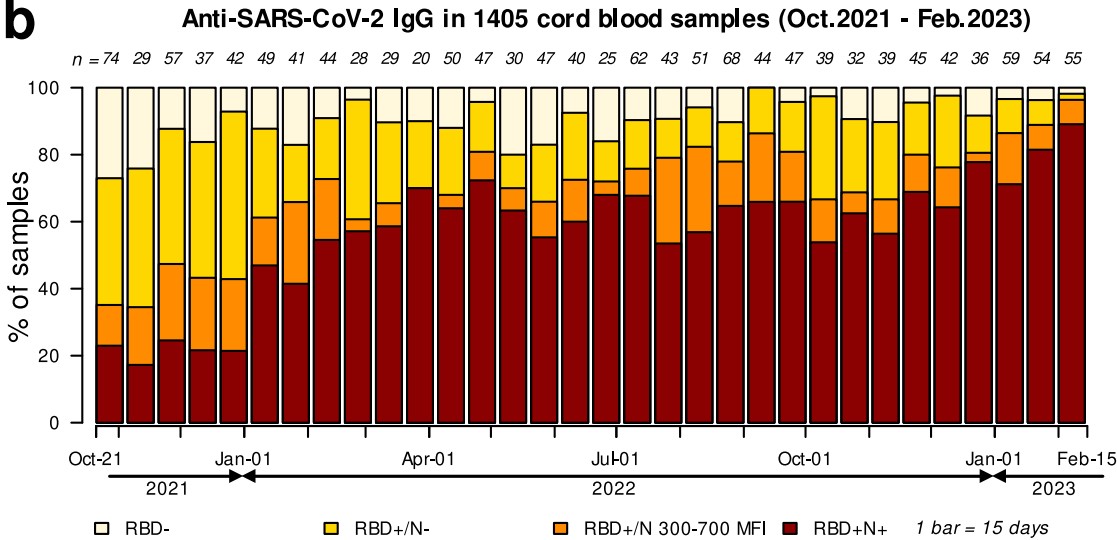

**b   Anti-SARS-CoV-2 IgG in 1405 cord blood samples (Oct.2021 - Feb.2023)**

RBD-   RBD+/N-   RBD+/N 300-700 MFI   RBD+N+   *1 bar = 15 days*

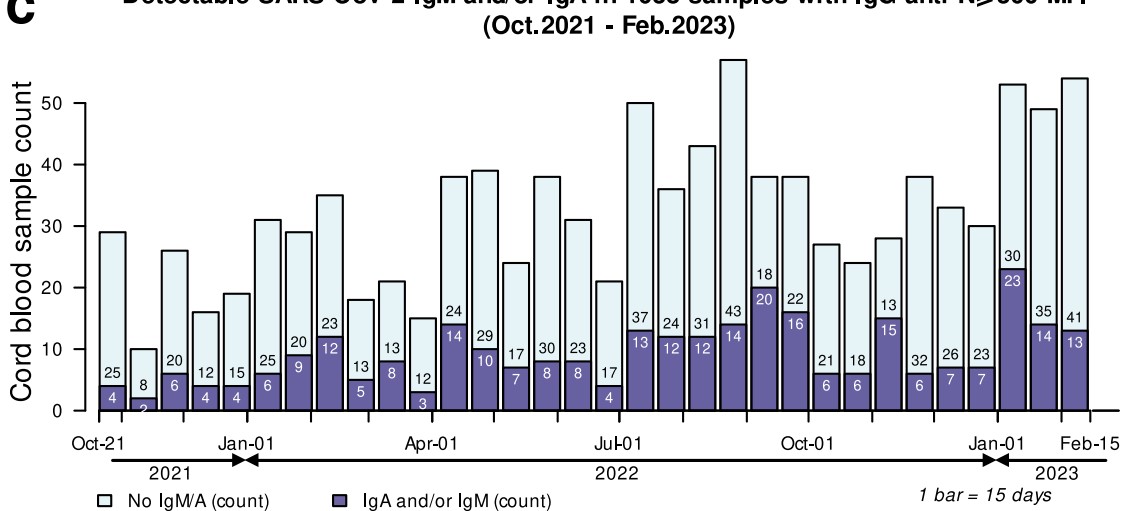

**c   Detectable SARS-CoV-2 IgM and/or IgA in 1038 samples with IgG anti-N≥300 MFI (Oct.2021 - Feb.2023)**

No IgM/A (count)   IgA and/or IgM (count)   *1 bar = 15 days*

Our study found that the prevalence of fetal serological responses to SARS-CoV-2 varied throughout the study period. There were distinct increased rates coincident with the new omicron sub-lineages circulating in the community. With serological evaluation over a 16-month period with multiple highly infectious waves and the evolution of multiple subvariants, we could detect changes in the prevalence of SARS-CoV-2 specific IgA/IgM antibodies suggesting potential in utero infection. For instance, the peaks observed with sub-lineages BA.1 (February 2022) and BA4/5 (September 2022) are especially marked after weeks or months of these variants circulating (Fig. 3), following a progressive increase in the rate of detectable IgA and IgM. These findings are consistent with infection late second and third trimester, during a window starting when the fetus can develop humoral response (especially IgA) and ending a few days before birth so IgM

**Fig. 1 | Serological profile of the Cord Blood Samples collected between October 2021 and February 2023. a** Evaluation of SARS-CoV-2 specific IgG in 1405 Cord Blood Samples (CBS). All CBS were tested for SARS-CoV-2 specific anti-RBD (Receptor Binding Domain) and anti-N (Nucleocapsid) IgG antibodies. Samples with anti-RBD above positivity threshold are represented in color: Samples with anti-RBD ≥700 MFI (Median Fluorescence Intensity) and anti-N < 300 MFI (bright yellow), anti-RBD ≥700 MFI and anti-N = 300–699 MFI (orange) and anti-RBD ≥700 MFI and anti-N ≥700 MFI (red). Source Data are provided as a Source Data File. **b** Anti-SARS-CoV-2 IgG in 1405 cord blood samples (CBS) collected between

October 2021 and February 2023. Biweekly count of anti-RBD (Receptor Binding Domain) and anti-N (Nucleocapsid) antibodies represented in color: CBS with anti-RBD < 700 MFI (Median Fluorescence Intensity) (light yellow), anti-RBD ≥700 MFI and anti-N < 300 MFI (bright yellow), anti-RBD ≥700 MFI and anti-N between 300–699 MFI (orange), and samples with anti-RBD ≥700 MFI and anti-N ≥700 MFI (red). Source Data are provided as a Source Data File. **c** Detectable SARS-CoV-2 IgA and/or IgM in 1038 cord blood samples (CBS) with IgG anti-N (Nucleocapsid) ≥300 MFI. Biweekly count of samples positive for anti-SARS-CoV-2 IgA or IgM (dark blue) versus negative for both (light blue). Source Data are provided as a Source Data File.

## Table 1 | Relationship between IgG anti-RBD (receptor binding domain) and anti-N (nucleocapsid) MFI (median fluorescence intensity) levels among cord blood samples with detectable SARS-CoV-2 IgA and/or IgM

| | Overall total tested for IgM/A N = 1038 | IgG anti-RBD MFI < 700 | | | IgG anti-RBD MFI ≥700 | | |
|---|---|---|---|---|---|---|---|
| | | IgG anti-N MFI | | Total tested for IgM/A N = 59 | IgG anti-N MFI | | Total tested for IgM/A N = 979 |
| | | 300–699 N = 18 | ≥700 N = 41 | | 300–699 N = 182 | ≥700 N = 797 | |
| Number with detectable IgM/A | 298 | 4 | 2 | 6 | 51 | 241 | 292 |
| Proportion with detectable IgM/A [95% CI] | 0.287 [0.260, 0.316] | 0.222 [0.064, 0.476] | 0.049 [0.006, 0.165] | 0.102 [0.038, 0.208] | 0.280 [0.216, 0.351] | 0.302 [0.271, 0.335] | 0.298 [0.270, 0.327] |
| Isotypes | | | | | | | |
| IgM+ n (%) | 31 (10.4) | 1 (25.0) | 1 (50.0) | 2 (33.3) | 10 (19.6) | 19 (7.9) | 29 (9.9) |
| IgM+ IgA+ n (%) | 40 (13.4) | 1 (25.0) | 1 (50.0) | 2 (33.3) | 8 (15.7) | 30 (12.4) | 38 (13.0) |
| IgA+ n (%) | 224 (75.2) | 2 (50.0) | 0 | 2 (33.3) | 33 (64.7) | 189 (78.4) | 222 (76.0) |
| IgA+ IgM unk n (%) | 3 (1.0) | 0 | 0 | 0 | 0 | 3 (1.3) | 3 (1.0) |
| P-value | 0.0003[c] | | | 0.0639[a] | | | 0.5908[b] |

All p-values were obtained by two-sided fisher's exact tests; 95% CIs were exact binomial CIs; IgM positive only denoted by 'IgM+'; both IgA and IgM positive denoted by 'IgA+ IgM+'; IgA positive only denoted by 'IgA+'; IgM positivity unknown denoted by 'IgM unk'.
Only samples with IgG anti-N ≥300 were tested for IgA and IgM.
[a]The difference in proportions with detectable IgM/A across N-levels among RBD/S <700.
[b]Testing the difference in proportions with detectable IgM/A across N-levels among RBD≥700.
[c]The difference in proportions with detectable IgM/A between RBD <700 vs. RBD≥700.

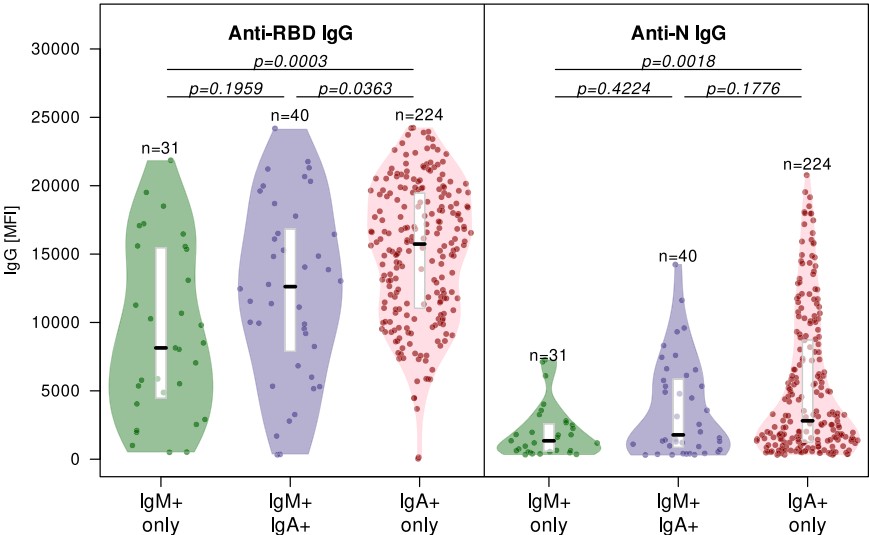

**Fig. 2 | Relationship between SARS-CoV-2 anti-RBD (receptor binding domain) and anti-N (nucleocapsid) specific IgG antibodies in all cord blood samples with detectable SARS-CoV-2 IgA and/or IgM, by isotype.** Isotypes were identified as IgM+ only (green, n = 31), IgA+ and IgM+ (purple, n = 40), and IgA+ only (red, n = 224). Three samples with unknown IgM are not included. The black bars represent the medians, and the white boxes represent the interquartile range.

Minima and maxima are represented by the base and top of the violins respectively. Wilcoxon rank-sum tests evaluated between group difference; Kruskal-Wallis test evaluated overall differences among the three groups. To account for multiple comparisons, Bonferroni correction was applied to pairwise p-values. The statistics used were all two-sided (for statistical analysis details, see Supplementary Tables S2 and S3). Source Data are provided as a Source Data File.

may be detectable. For example, newborns born at the beginning of the BA.4/5 wave are more likely to have been infected with BA.2, while newborns born at the end of the BA.4/5 wave are more likely to have been infected by BA.4/5.

Two comprehensive systematic reviews and meta-analyses estimated the potential for in utero infection[5,6]. Rates of reported in utero infection have ranged from 0 to 9.6%[1,5,6,9]. However, most of these studies were small or tested only for IgM or SARS-CoV-2 infection at

**Table 2 | Monthly estimates of cord blood samples with detectable SARS-CoV-2 IgA and/or IgM (N = 1038)**

| Month | No. +IgM/A | Total N | Estimate[a] | Wald 95% Confidence Intervals | |
|---|---|---|---|---|---|
| Oct-21 | 4 | 18 | 0.222 | 0.030 | 0.414 |
| Nov-21 | 6 | 37 | 0.162 | 0.043 | 0.281 |
| Dec-21 | 10 | 40 | 0.250 | 0.116 | 0.384 |
| Jan-22 | 14 | 62 | 0.226 | 0.122 | 0.330 |
| Feb-22 | 18 | 50 | 0.360 | 0.227 | 0.493 |
| Mar-22 | 11 | 41 | 0.268 | 0.133 | 0.404 |
| Apr-22 | 20 | 62 | 0.323 | 0.206 | 0.439 |
| May-22 | 18 | 73 | 0.247 | 0.148 | 0.345 |
| Jun-22 | 13 | 57 | 0.228 | 0.119 | 0.337 |
| Jul-22 | 21 | 76 | 0.276 | 0.176 | 0.377 |
| Aug-22 | 27 | 105 | 0.257 | 0.174 | 0.341 |
| Sep-22 | 39 | 80 | 0.488 | 0.378 | 0.597 |
| Oct-22 | 12 | 49 | 0.245 | 0.124 | 0.365 |
| Nov-22 | 21 | 66 | 0.318 | 0.206 | 0.431 |
| Dec-22 | 14 | 66 | 0.212 | 0.113 | 0.311 |
| Jan-23 | 40 | 108 | 0.370 | 0.279 | 0.461 |
| Feb-23 | 10 | 48 | 0.208 | 0.093 | 0.323 |

[a]Estimate: The proportion with detectable SARS-CoV-2 IgA and/or IgM.

birth using PCR[5,6,9,18,23]. Testing only for infection by PCR has challenges as the virus can clear rapidly and may not be detected by PCR at birth if infected earlier in pregnancy[9], making the diagnosis of in utero infection more difficult.

IgM antibodies are the first to develop after exposure to a new antigen, and then levels decrease while the more specific and longer lasting IgA rises. If the infection occurs weeks before delivery, IgM might not be detectable at time of birth, but IgA may likely still be present. If the infection occurs in the days before delivery, IgA might not yet be detectable in cord blood. Multi-isotype panels that include IgA have been used to diagnose other congenital infections, such as Zika Virus[24], toxoplasmosis[24–26] and Rubella[24,27]. Interestingly, a study focusing on SARS-CoV-2 confirmed that vaccine induced IgG crossed the placental barrier[15]. While the authors could not detect IgM transfer after vaccination, they did detect anti-SARS-CoV-2 IgM in 5 newborns, all related to recent maternal COVID infection[15].

It is generally accepted that maternal IgG antibodies are actively transported across the placenta via placental Fc receptors[16–18], while IgM and IgA antibodies found in the cord blood originate from the fetus[8,16,21,28]. Evidence suggests the fetus begins making specific IgA antibodies at 24–27 weeks[26,29]. However, no large study has evaluated SARS-CoV-2 specific IgA and IgM antibodies in newborns. Our study found a significant association between SARS-CoV-2 specific IgG levels and presence of newborn IgM and IgA. Median anti-RBD IgG and anti-N IgG MFI levels were significantly lower among those with IgM only compared to those with IgA only, suggesting more recent infection. This can be linked to the temporal relationship between the different isotypes and further supports that IgA and IgM antibodies are specific indicators of in utero infection, as recent studies confirmed that IgG levels were lower when infection was closer to or at delivery[21].

The RBD is located on the Spike glycoprotein and can therefore be targeted by neutralizing antibodies. IgA anti-RBD is the predominant neutralizing antibody[12,21] and provides some protection across variants[30,31]. Its production may be critical for protection of the fetus[32]. Neutralizing IgA antibodies are efficiently transferred during

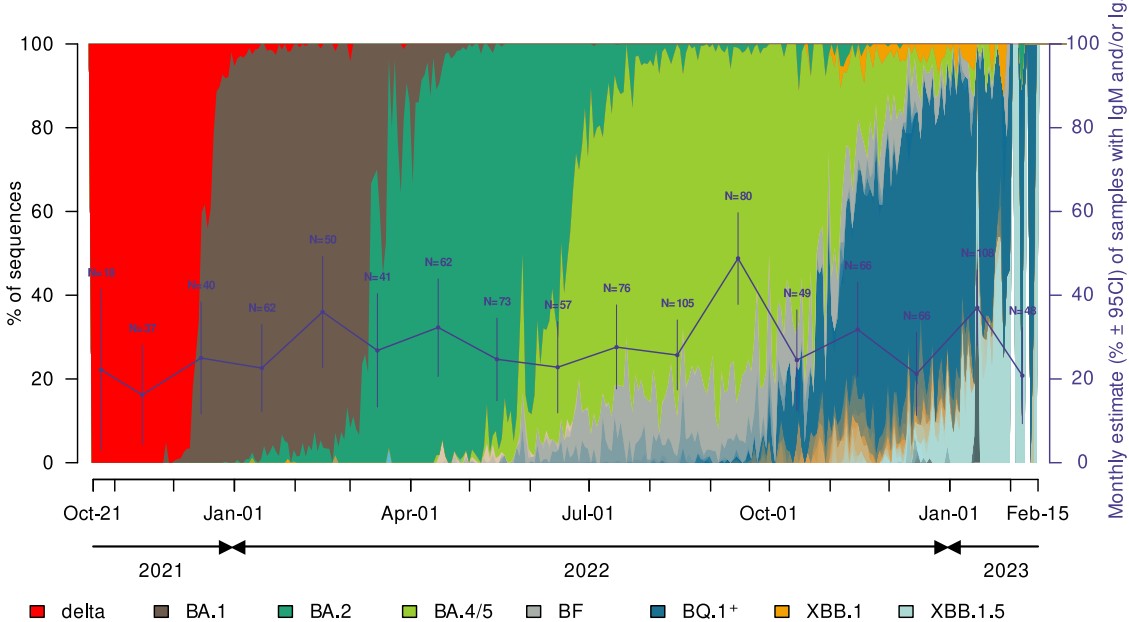

**Fig. 3 | Cord blood IgM and IgA responses to SARS-CoV-2 in relation to viral lineage evolution (Oct. 2021–Feb. 2023).** Proportions of variants and lineages sequence analyses publicly available in the GISAID database for Los Angeles County during the study period: https://doi.org/10.55876/gis8.231023qv. Monthly variations in prevalence rate of cord blood samples with detectable SARS-CoV-2 IgA and/ or IgM over the study period is represented by the dark blue line (for details, see also Table 2). Vertical lines at the different months represent Wald 95% confidence intervals, and the N is the total sample size for the month. +BQ.1 combines BQ.1, BQ.1.1, and BQ.1.2.

breastfeeding[30–32], and likely help control perinatal infection[21,30–32]. While IgA has been shown to persist up to 8 months after infection in adults[11], fetal data are unavailable. IgA antibodies produced by the fetus at around 24–27 weeks[26,29] may persist for weeks to months and relocate to mucosal surfaces, providing mucosal protection against future infection, including from different variants[30,31].

Surprisingly, we found that screening newborn CBS for anti-N IgG antibodies alone was insufficient to identify potential in utero infection since some had IgG anti-N antibodies below positivity threshold (Table 1). While 97.9% of the CBS with IgA/IgM specific for SARS-CoV-2 had IgG anti-RBD levels above the assay threshold (≥700 MFI), only 81.5% had anti-N IgG levels above threshold. These results confirm that anti-N levels vary considerably between patients depending on factors such as variant type, vaccination status, maternal symptomatology, and decline of anti-N over time after infection[21,33]. Maternal IgG anti-N might also not be present at delivery if primary infection is recent[22].

Our study has the strength of a large cohort that included 90% of newborns born at a single hospital representing the same population over more than 1 year. This allowed us to evaluate the prevalence of fetal response to SARS-CoV-2 as different variants emerged. However, there are limitations. First, this study does not include samples from the first waves of the pandemic. We evaluated serological markers of potential in utero infection during a unique period when the Delta and Omicron variants emerged and infected much of Los Angeles County population (highlighted in Fig. 3) despite high rates of vaccination[34]. Mutations in the Spike glycoprotein observed across different variants might have impacted viral dynamics and abilities to infect and cross the placenta[6,9,16]. Second, while our assay is specific and was validated with pre-2019 newborn CBS, results may differ depending on circulating variants, as the assay targets antigens that are based on the original sequence for RBD and N. Thus, antibodies specific to peptides unique to a new variant may not have been detected. Further, we used a very conservative threshold for antibodies with a false positive rate approaching zero[35]. Along with strict internal controls and IgG depletion, we further limited false positivity due to isotype cross-reactivity and false negative results due to IgG competition for the epitopes. However, this overly conservative approach might have led to underestimation of potential in utero infection in general. For example, if the fetus was infected just prior to birth, SARS-CoV-2 specific IgM, IgA, and IgG antibodies may not yet be detectable at birth[22]. This is why clinically validated assays typically require serial positive results, and the WHO definition of SARS-CoV-2 vertical transmission requires confirmatory testing using a second specimen[8]. Maternal blood contamination of fetal cord blood can happen during late pregnancy and delivery through maternal fetal transfusion, and also during cord blood collection. Studies have shown that contamination occurs between 0 and 22% of the samples, with wide variation between studies, depending on collection and analysis methods[36,37]. However, contaminating material only represents a fraction of the sample, with dilution ranging from $10^5$ to $10^{4\ 36}$. In our study, potential contamination was minimized as cord blood was collected using the UmbiliCup, which minimizes maternal blood contamination.

In summary, 28.7% of CBS from neonates born to mothers with suspected past infection, had SARS-CoV-2 specific IgA or IgM antibodies suggesting in utero infection, with distinct variations in prevalence over the study period as new variants emerged. Anti-RBD IgA antibodies, which are strongly neutralizing, were the predominant isotype found and may be important in protecting the fetus and newborn. Our study demonstrates that in utero infection may have previously been underestimated, most likely due to the almost exclusive use of IgM and/or positive PCR as in utero infection indicators. Screening neonatal cord blood for SARS-CoV-2 specific IgA along with IgG and IgM antibodies can be a reliable strategy to identify newborns with serological markers of potential in utero infection so that they can be followed more closely and evaluated over time for infant outcomes.

## Methods

### Cord blood samples
The University of Southern California/Health Science Institutional Review Board (IRB) approved the study protocol as exempt, as samples were deidentified and only remnant samples that were to be discarded were used. Between October 21, 2021, and February 15, 2023, all available remnant CBS from neonates born at LA General were obtained from the hospital Blood Bank prior to disposal. CBS are routinely collected from all newborns and stored for 8 days before being discarded. Cord blood is collected using the UmbiliCup (DeRoyal, Powell, TN, USA) cord blood collection device, which is a sterile container for collecting umbilical cord blood. Following the manufacturer guidelines the cord is clamped in two places with the first clamp placed near the placenta, and the second one on the newborn side. The UmbiliCup is opened, and the cord blood is collected by slowly releasing the second clamp. An empty vacutainer tube is then inserted into the UmbiliCup needle sleeve so that there is minimal blood exposure risk to the collector and the collection tube is never exposed to the maternal blood. During the study period, there were 1561 live births at the hospital, of which 1405 (90%) remnant CBS were available for serological testing.

We also tested 103 CBS, collected before 2019 as part of an IRB approved ongoing study, as true negative controls.

### Sample processing
EDTA and no additive (NON) remnant cord blood samples are held at 2–8 °C in the LA General Medical Center Blood Bank for approximately 8 days post-delivery. The specimens were coded and transported in temperature-controlled biohazard bags to our laboratory (the Maternal, Child/Adolescent Center for Infectious Diseases and Virology Laboratory) for processing. EDTA and no additive tube types were centrifuged at 900–1000 g for 10 min. Serum and plasma were aliquoted into 2 equal aliquots and frozen at −70 °C (−55 °C to −85 °C). Specimens were thawed at 4 °C prior to being assayed.

### Methods for anti-SARS-CoV-2 IgG antibodies
We used the Luminex xMAP® SARS-CoV-2 Multi-Antigen Antibody Assay (Luminex – Austin, TX, USA) to semi-quantitively determine anti-RBD and anti-N IgG antibody levels following manufacturer recommended protocols and previous reports[13,19]. Fluorescence was read using Luminex MagPix controlled with the xPonent software (Luminex, Austin, TX, USA), reported as Median Fluorescence Intensity (MFI), and analyzed with R v4.3.3 (R Foundation for Statistical Computing, Vienna, Austria). Limit of Blank (LoB) and Limit of Detection (LoD) for IgG, as well as the specificity and cross reactivity with other coronaviruses, were determined by the manufacturer[19]. Each run included control samples from the WHO IgG Std – EN63QG – 20/136 (World Health Organization), Luminex xMAP SARS-CoV-2 Controls (Luminex, Austin, TX, USA), and Bio-Plex Pro Human IgG SARS-CoV-2 Controls (Bio-Rad, Hercules, CA, USA). As part of quality control prior to testing we compared antibody results of fresh cord blood and 8-day old cord blood, finding comparable results. Assay validation included running randomly selected samples multiple times and WHO standards within each run to ensure reproducibility.

### Methods for anti-SARS-CoV-2 IgA and IgM antibodies
Since newborn specific SARS-CoV-2 IgM, IgA and IgG antigen-specific antibodies develop sequentially and may decline overtime, we tested all samples with IgG anti-N levels above background (≥300 MFI) for IgA/IgM. We used a modified protocol recommended by the manufacturer[19,20] that included depletion of the IgG with biotinylated goat anti-Human IgG (ab97223; Abcam) bound to Streptavidin Magnetic Particles (11641786001; Roche/Millipore-Sigma, Burlington, MA, USA). IgG detection reagents were replaced with 1:1600 anti-human IgA coupled with r-phycoerythrin, and anti-human IgM coupled with

r-phycoerythrin (109-115-011, 109-116-129 respectively; ImmunoResearch). All IgG depleted samples were shown to have IgG below background levels prior to IgM and IgA testing.

### Establishing LoD for SARS-CoV-2 IgA and IgM antibody testing

Using 103 neonatal CBS collected before 2019, expected to be true negatives, we calculated the LoD for IgM and IgA (Supplementary Fig. S1). The LoD is usually defined as $LoB + 1.645 \times SD_{blank}$ where LoB is the highest apparent analyte concentration of a negative sample $(mean_{blank} + 1.645 \times SD_{blank})$[35]. However, we set the LoD at $LoB + 5 \times SD_{blank}$ to ensure a conservative estimation and optimize test specificity. We determined IgM anti-RBD limit threshold to be $\geqslant 45.5$ MFI and anti-N $\geqslant 78.37$ MFI. IgA anti-RBD threshold was $\geqslant 10.63$ MFI and anti-N $\geqslant 11.96$ MFI (Supplementary Table S1). Evidence of in utero infection was defined as anti-RBD or anti-N IgA or IgM antibodies above these defined thresholds. If both anti-RBD and anti-N IgA and IgM levels were below these thresholds, newborns were considered not to have serological evidence of in utero infection.

### Establishing specificity for SARS-CoV-2 IgA and IgM antibody testing

To ascertain the specificity of the IgA/IgM assay, we performed a Monte Carlo simulation with the gamma distribution to fit the non-normal distribution of IgA and IgM MFI levels from the 103 pre-2019 neonatal cord blood specimens. Using the traditional threshold $(LoB + 1.645 \times SDblank)$, the false positive rate was 0.52% (95% CI: 0.48%, 0.57%) for IgA anti-RBD, 0.64% (95% CI: 0.61%, 0.71%) for IgA anti-N, 1.03% (0.97%, 1.09%) for IgM anti-RBD, and 0.71% (95% CI: 0.67%, 0.77%) for IgM anti-N. Adopting the more conservative threshold $(LoB+5 \times SDblank)$ reduced false positives to 0.001% (95% CI: 0%, 0.004%) for IgA anti-RBD, 0.003% (95% CI: 0%, 0.006%) for IgA anti-N, 0.016% (95% CI: 0.009%, 0.025%) for IgM anti-RBD, and 0.004% (95% CI: 0.001%, 0.008%) for IgM anti-N in the simulation results (Supplementary Table S1). The probability of falsely detecting SARS-CoV-2 antibodies in pre-pandemic samples is exceedingly low, enhancing the specificity of the testing method.

### Variant timeline

To correlate rates of fetal IgA/IgM over time with circulating SARS-CoV-2 variants in the community, we used all available SARS-CoV-2 sequence analyses ($n = 29,487$) publicly available in the GISAID database for Los Angeles County between October 21, 2021 to February 15, 2023 (dataset EPI_SET_231023qv – https://doi.org/10.55876/gis8.231023qv).

### Statistical analyses

The proportion of samples with detectable SARS-CoV-2 IgA and/or IgM and 95% confidence intervals (95% CI) were calculated for each category of anti-RBD and anti-N level. Fisher's exact tests evaluated differences in the proportions with detectable IgM/A between 2 anti-RBD strata and within each anti-N strata (Table 1).

Median IgG anti-RBD levels and IgG anti-N levels were compared among three isotype groups: samples IgM positive only, IgA positive only, and both IgM and IgA positive. Wilcoxon rank-sum tests evaluated between group difference and Kruskal-Wallis tests evaluated overall differences among the three groups. To account for multiple comparisons, Bonferroni correction was applied to pairwise $p$-values (Supplementary Tables S2 and S3).

To test the probability of in utero detectable SARS-CoV-2 IgA and/or IgM antibodies varying by month (October 2021 to February 2023), a type III test was performed in generalized linear regression using a binomial distribution with an identity link function. $P$-value of <0.05 was considered statistically significant.

The statistics used were all two-sided. All analyses were conducted using SAS 9.4.

### Reporting summary

Further information on research design is available in the Nature Portfolio Reporting Summary linked to this article.

## Data availability

Source Data are provided with this paper. SARS-CoV-2 sequence for Los Angeles County between October 21, 2021 to February 15, 2023 ($n = 29,487$) are publicly available in the GISAID database (dataset EPI_SET_231023qv – https://doi.org/10.55876/gis8.231023qv). Source data are provided with this paper.

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

## Acknowledgements

This work was supported by grants UL1TR001855 and UL1TR000130 from the National Center for Advancing Translational Science (NCATS) of the U.S. National Institutes of Health. The content is solely the responsibility of the authors and does not necessarily represent the official views of the National Institutes of Health. This study was also supported by NIH, NIAID 1R56AI178166-01A1; Andrea Kovacs M.D. Principal Investigator.

## Author contributions

O.P., A.K. designed the study, analyzed the data, and wrote the manuscript. O.P. generated the figures and tables. T.F. wrote the manuscript, analyzed the data and generated the tables. A.A. and W.J.M. performed the statistical analysis, generated the tables, and participated in the manuscript preparation and review. P.A., G.M. performed the assays, blood processing and wrote the manuscript. S.B. performed the assays and blood processing. E.N., J.L. performed the data management. A.K., J.H., M.B., A.Y., A.B., R.R. developed the clinical protocol to obtain and code specimens and participated in the manuscript preparation and review. Raw data have been reviewed by O.P., P.A., and A.K.

## Competing interests

The authors declare no competing interests.
