## [Peer Review file · Nature Communications]

Cord Blood IgA/M Reveals In Utero Response to SARS-CoV-2 with Fluctuations in Relation to Circulating Variants

Corresponding Author: Professor Olivier Pernet

Version 0:

Reviewer comments:

Reviewer #1

(Remarks to the Author)

This is an interesting manuscript which reports potentially higher rates of in utero SARS CoV-2 transmission through the testing of cord blood for SARS CoV-2 specific IgM and/ or IGA in 1038 cord blood specimens available. Pernet et al. present compelling data on fluctuating SARS-CoV-2 cord blood IgM and/or IgA levels in relation to circulating variants at the time of collection. Using remnant cord blood samples collected from neonates from a single hospital in Los Angeles from October 2021 to February 2023, SARS-CoV-2 fetal IgM and/or IgA were detected in 28.7% of 1038 cord blood samples tested. The most striking peak was during the BA.4/5 wave in September 2022, with 48.8% of samples testing positive.

All newborn specimens were screened for IgG anti-Receptor Binding Domain (RBD) antibodies, which are part of the spike glycoprotein present in natural infection or vaccines and anti-nucleocapsid (N) antibodies which are present following natural infection. When IgG specimens were present for both antibodies, which is indicative of past maternal infection, cord blood specimens were evaluated for IgA and IgM antibodies targeting RBD and N proteins. Specimens consisted of de-identified remnant cord blood samples available at the institutional blood bank, with 1405 specimens from live born infants available for testing, in addition to 103 specimens from prior to the pandemic as negative controls. In total, 90% of specimens were positive for anti-RBD while 57% were positive for antibodies against RBD and the N protein. High rates of IgG positivity were noted during the pandemic, progressing to 80% by the end of the Omicron wave. Seventy-four percent of specimens had detectable IgG N antibodies above the threshold and were thus tested for IgA and IgM. Either or both antibody isotypes were present in 28.7% of cord blood specimens. Detection of positive IgM or IgA in cord blood specimens varied over time depending on the circulating variants.

The manuscript makes a compelling argument for a much higher vertical transmission rate of SARS CoV-2 based on fetal-specific humoral responses than originally reported (about 27% overall but ranging from 16.2% to 48.8% depending on the timing of the pandemic). Most studies have suggested vertical transmission rates ranging from 2 to 4%, but testing was definitely not as extensive in most reports. The methodology is sound and the testing of both IgM and IgA-specific antibodies against two viral targets increases the sensitivity and specificity.

One question that comes to mind, however, which would be important for the authors to address is the possibility of cord blood contamination with maternal blood, which would obviously influence results. How frequently does cord blood contamination with maternal blood occur, and what would be the best approach to address this issue? Minimally it should be mentioned as a study limitation, with a few sentences about this possibility welcomed in the discussion. While the team demonstrates high levels of IgM and/or IgA in cord blood samples, it is unclear if the presence of IgM and/or IgA in the absence of a second, confirmatory test represents true in utero transmission. Cord blood is not a perfect surrogate for peripheral infant blood, as umbilical cord blood may contain maternal blood, either by cross-contamination during sample collection, or more commonly, from maternal blood entering fetal circulation during labor. Per the WHO definition of vertical transmission of SARS-CoV-2, "a positive serological test requires confirmatory testing of a second specimen, preferably using molecular diagnostic tests to directly detect the pathogen or otherwise a later serological test." While PCR alone is unreliable for diagnosis in neonates, PCR positivity from another site paired with positive cord blood serologies is highly suggestive of vertical transmission. Therefore, whether anti-RBD or anti-N IgM and/or IgA collected from cord blood represents true in utero SARS-CoV-2 infection (IU-SC) remains unclear.

Overall, the methods are sound, the data are compelling, and the paper is very clear and well-written. The research team should address the issue of the use of cord blood specimens to measure vertical transmission. The approach may include a

change in the title and focus of the paper from in utero transmission, and instead focus on the fetal response to in utero infection, which is well-supported by the current data. They may review the literature and try to tease out how often cord blood is contaminated with maternal blood, in order to potentially statistically correct for this possibility. Definitely the use of cord blood for determination of in utero infection rates should be highlighted as a study limitation.

Other comments:

Background

Line 117- should likely be RBD and N proteins or antigens, not antibodies.

As mentioned above, it is unclear if the presence of cord blood IgM and/or IgA in the absence of confirmatory tests represents newborn IU-SC. Therefore, the "prevalence of newborn IU-SC" (line 122) cannot be determined based on this definition, but rather the evidence of the fetal response to maternal infection and/or maternal IgM and IgA responses to infection.

Methods

In the Cord Blood Samples section, were blood separated into plasma or sera? Were they stored at -80 deg C?

Results

Line 237- would provide the percentage of positives (74% for 1038/1405).

Discussion

Line 276 notes that "no published studies have assessed for both fetal IgM and IgA SARS-CoV-2 specific antibodies to diagnose IU-SC in a large cohort." However, would note that technically fetal IgM and IgA were not measured, as these were cord blood samples.

Reviewer #2

(Remarks to the Author)

The authors examined neonatal cord blood samples for SARS-CoV-2 specific IgA and IgM antibodies, and they correlated these serologic findings with epidemiological data on circulating COVID-19 variants. They suggest that the presence of SARS-CoV-2 specific IgA and/or IgM antibodies in neonatal cord blood is indicative of in utero infection. In their study, 28.7% of samples were positive for IgA and/or IgM antibodies, which is higher than previous estimates of in utero transmission based on PCR and IgM antibodies.

While the detection of SARS-CoV-2 specific IgA antibodies in neonatal cord blood suggest possible in utero infection, there are limitations to the use of IgA antibody testing that should be addressed. The authors should provide more commentary on the use of cord blood IgA to diagnose vertical transmission of other infections and the false positive rate of IgA antibodies. Diagnosis of SARS-CoV-2 fetal infection by IgM testing typically requires serial positive results, and it is unknown whether a single positive IgA antibody result in neonatal cord blood is sufficient to make the diagnosis of fetal infection. Although IgA antibodies are worth exploring as markers for in utero transmission, the conclusions drawn from this study should be moderated.

Additionally, the authors rely on epidemiological data on circulating variants in Los Angeles to determine variant-specific risk of in utero infection. However, use of variant-specific SARS-CoV-2 antibody testing would be a more reliable way to answer this question.

Specific Comments:

1. Lines 132-140: Can the authors expand on their protocol for cord blood collection? Were samples collected via umbilical venipuncture? How can they be sure there was not maternal blood contamination?
2. Lines 132-140: What were the reasons for not being able to collect 10% of the cord blood samples?
3. Lines 153-162: Please provide more details on the experimental methodology. For instance, was plasma or serum tested? Were all experiments conducted simultaneously (and if so when), or were experiments conducted serially as cord blood samples were collected? How did the authors control for batch effects? What was the sample dilution? An inappropriately high dilution can contribute to false positive results. Did the authors run samples in duplicate/triplicate?
4. Was there any consideration to performing variant-specific SARS-CoV-2 IgA/IgM antibody testing? This would strengthen the study's findings regarding variant-specific risk of in utero transmission.
5. It would be helpful if the authors could comment more on the use of IgA antibodies as an indicator of fetal infection. Are there other infections that use cord blood IgA antibodies to diagnose fetal infection? At what gestational age do fetuses begin making IgA antibodies in response to fetal infection?
6. Although we understand that the cord blood samples are deidentified, were the authors able to collect any clinical information on these patients, such as fetal sex or gestational age at delivery? If so, this would be helpful to include in the manuscript.
7. What is the false positive rate of IgA antibodies? IgM antibodies are known to have high false positive rates, and diagnosis of in utero SARS-CoV-2 infection generally requires multiple positive IgM tests collected over time with a pattern that serially declines.
8. What it be possible to confirm how many patients that were IgA/IgM antibody positive were also PCR positive?
9. Lines 223-224: It is not necessarily true that anti-RBD and anti-N IgG antibodies are indicative of maternal infection during

pregnancy. Maternal IgG antibodies persist indefinitely and cross the placenta, thus the presence of anti-RBD and anti-N IgG antibodies may be due to a remote infection prior to pregnancy.

10. Are the authors able to comment on the impact of maternal vaccination status on the results?

11. Lines 282-283: Although the presence of SARS-CoV-2 specific IgA antibodies in neonatal cord blood samples is suggestive of in utero infection, we do not feel that the results of this study are sufficient to conclude that in utero infection has occurred. The study is limited by having a single isolated sample (rather than serial infant samples), the potential for false positive IgA/IgM results, the possibility of maternal blood contamination in cord blood samples, and the absence of PCR data to confirm in utero infection. We agree that IgA antibodies should be explored in future work as a potential marker for in utero transmission, but we feel the conclusions should be softened as this study does not provide sufficient evidence of in utero COVID-19 infection.

Reviewer #3

(Remarks to the Author)

Reviewer #4

(Remarks to the Author)

Version 1:

Reviewer comments:

Reviewer #1

(Remarks to the Author)

The authors have adequately addressed our concerns, including expanding the methods, using more precise language, and incorporating our concerns regarding study limitations in the discussion. The manuscript is very clear, and the study was well-executed.

Reviewer #2

(Remarks to the Author)

This is a revised manuscript describing the detection of SARS-CoV-2 IgA/IgM antibodies in cord blood samples as an indicator of possible in utero transmission. The authors have generally responded to the prior comments, specifically by including details regarding sample collection methods, as well as discussion of the accuracy of IgA testing in cord blood. While the conclusions of the manuscript appear to have been tempered in response to prior reviews, and a couple additional references have been included regarding IgA testing for in utero transmission, I feel that the emphasis is still too strong that IgA positivity in cord blood is indicative of in utero transmission. I agree with the prior reviewer that it is indicative of fetal response to maternal infection, as viral proteins or antigens (not infectious virus) may cross the placental barrier and stimulate the fetal immune response.

Furthermore, testing of a single sample and not in duplicate, the experimental rigor is reduced. The vast majority of Luminex bead-based assays are conducted in duplicate or even triplicate, per the manufacturer's recommendations. For a first study reporting on a novel antibody detection method, technical replicates are critical to ascertain the precision and reproducibility of the assay. Running samples just once may be acceptable in a well-established assay, but the novelty of this study is the new assay.

Reviewer #3

(Remarks to the Author)

Here is a point-by-point response to the Reviewer's comments:

REVIEWER COMMENTS

Reviewer #1 (Remarks to the Author):

Critique: One question that comes to mind, however, which would be important for the authors to address is the possibility of cord blood contamination with maternal blood, which would obviously influence results. How frequently does cord blood contamination with maternal blood occur, and what would be the best approach to address this issue? Minimally it should be mentioned as a study limitation, with a few sentences about this possibility welcomed in the discussion. While the team demonstrates high levels of IgM and/or IgA in cord blood samples.

Response: We describe the protocol for cord blood collection which uses the *Umbilicup*, a sterile device that minimizes contamination. We also further reviewed the literature regarding cord blood collection procedures and associated risk, frequency, and average volume of maternal blood contamination. We used these parameters to detail the limitations of our study. We conclude that maternal blood contamination would have a minimal and non-statistically significant impact on the overall conclusions in this study, because we used highly conservative positivity thresholds with extremely low false positivity rates.

Other comments:

1. Background

Line 117- should likely be RBD and N proteins or antigens, not antibodies.

Response: This typo has been corrected.

2. As mentioned above, it is unclear if the presence of cord blood IgM and/or IgA in the absence of confirmatory tests represents newborn IU-SC. Therefore, the "prevalence of newborn IU-SC" (line 122) cannot be determined based on this definition, but rather the evidence of the fetal response to maternal infection and/or maternal IgM and IgA responses to infection.

Response: Unlike IgG, SARS-CoV-2 specific IgM and IgA in cord blood are of fetal origin as they do not cross the placenta, and "*represent fetal immune response to infection*" (WHO guidelines). This has been confirmed for SARS-CoV-2 antibodies (see references #15 and #29 for examples). With the *Umbilicup* device and collection protocol used at LA General Hospital, it is less likely that samples were contaminated by maternal IgA or IgM. Further, if there were contamination, it would have to be a major amount of maternal blood entering the fetal circulation. We explain why potential maternal contamination would not have impacted our results. We agree that presence of cord blood IgM and/or IgA without confirmatory tests is not sufficient to diagnose IU-SC, and have changed sentences stating, "prevalence of newborn IU-SC" to "evidence of IU-SC". We also replaced "indicating IU-SC" with

“suggesting IU-SC” throughout the manuscript. However, we do feel that it is essential to have a screening test that maximally identifies newborns who may have been infected in utero so that they can be followed for further testing and evaluation for any negative outcomes.

3. Methods

In the Cord Blood Samples section, were blood separated into plasma or sera? Were they stored at -80 deg C?

Response: A *Sample Processing* section has been added to the *Methods*. Both Serum and Plasma were collected, aliquoted into two equal aliquots, and frozen at -70°C (-55°C to -85°C). Specimens were thawed at 4°C prior to being assayed.

We ran sera with few exceptions. When sera were not available, we ran plasma. The validation process has not shown significant differences between sera and plasma, and the manufacturing company for the assay lists both sample types as acceptable.

4. Results: Line 237- would provide the percentage of positives (74% for 1038/1405).

Response: The percentage (74%) has now been included.

5. Discussion: Line 276 notes that “no published studies have assessed for both fetal IgM and IgA SARS-CoV-2 specific antibodies to diagnose IU-SC in a large cohort.” However, would note that technically fetal IgM and IgA were not measured, as these were cord blood samples.

Response: The text has been edited to replace “fetal” by “cord blood.”

Although cord blood is fetal blood, we did not do fetal sampling prior to birth. Cord blood represents blood from the fetal side of the placenta. Unlike IgG, IgM and IgA in cord blood are of fetal origins as multimeric isotypes do not cross the placenta.

Reviewer #2 (Remarks to the Author):

While the detection of SARS-CoV-2 specific IgA antibodies in neonatal cord blood suggest possible in utero infection, there are limitations to the use of IgA antibody testing that should be addressed. The authors should provide more commentary on the use of cord blood IgA to diagnose vertical transmission of other infections and the false positive rate of IgA antibodies. Diagnosis of SARS-CoV-2 fetal infection by IgM testing typically requires serial positive results, and it is unknown whether a single positive IgA antibody result in neonatal cord blood is sufficient to make the diagnosis of fetal infection. Although IgA antibodies are worth exploring as markers for in utero transmission, the conclusions drawn from this study should be moderated.

Response: Cord blood testing for IgA has been utilized for years for toxoplasmosis and other newborn TORCH infections (see references #25-28 for examples) as we review in the discussion section. We believe it is essential to screen all newborns for potential SARS-CoV-2 infection. Using cord blood is the most practical and accessible way to screen all births. Since cord blood is routinely used for genetic testing and is rich in stem cells that can be stored, maternal contamination in cord blood samples has

been studied by others. We added references #36 and #37 to illustrate and discuss this issue. With the use of the *Umbilicup*, which is a sterile device, contamination is minimized.

Additionally, the authors rely on epidemiological data on circulating variants in Los Angeles to determine variant-specific risk of in utero infection. However, use of variant-specific SARS-CoV-2 antibody testing would be a more reliable way to answer this question.

Response: Variant specific antibody testing is not readily available and would not be feasible for large studies where the variants are changing so rapidly. The importance of our study is to have an assay that can detect fetal antibody response at birth so that these infants can be more closely followed with repeat testing and evaluation for developmental abnormalities and other abnormalities that may be associated with in utero infection. Currently, there is no routine way of screening all births for potential SARS-CoV-2 infection.

Specific Comments:

1. Lines 132-140: Can the authors expand on their protocol for cord blood collection? Were samples collected via umbilical venipuncture? How can they be sure there was not maternal blood contamination?

Response: All cord blood samples are collected using the *Umbilicup* device, following the manufacturer guidelines. Briefly, a first clamp is placed near the placenta, and another one on the newborn side. The *Umbilicup* is opened, and the cord blood is collected by slowly releasing the second clamp. An empty vacuum tube is then inserted into the *Umbilicup* needle sleeve. That way, the collection tube is never exposed to the maternal blood. This explanation has been added to the methods section.

2. Lines 132-140: What were the reasons for not being able to collect 10% of the cord blood samples?

Response: It is possible that cord blood samples were not collected, or they were collected but not available as they can be used for newborn diagnostics/care. Only samples with leftover volume available after 8 days were used in this study. Cord blood is routinely collected and stored on all babies and if needed to type and crossmatch, or for some other reason, it may not have been available for us to use.

3. Lines 153-162: Please provide more details on the experimental methodology. For instance, was plasma or serum tested? Were all experiments conducted simultaneously (and if so when), or were experiments conducted serially as cord blood samples were collected? How did the authors control for batch effects? What was the sample dilution? An inappropriately high dilution can contribute to false positive results. Did the authors run samples in duplicate/triplicate?

Response: We added a *Sample Processing* section in the Methods.

Both Serum and Plasma were collected, aliquoted into 2 equal aliquots, and frozen at -70°C (-55°C to -85°C). Specimens were thawed at 4°C prior to being assayed. We ran sera with few exceptions. When

serum was not available, we ran plasma. The validation process has not shown significant differences between sera and plasma.

Experiments were conducted serially as cord blood samples were collected. We control for batch effects by using a serial dilution of the WHO standard, as well as 2 other commercial controls (Bio-Rad, Luminex). Additionally, the manufacturer provided its own lot-specific control datasheet.

Samples were typically run only once if they passed all quality controls. Unlike ELISA, this is a bead-based assay which requires a minimum number of beads per sample increasing the surface area significantly. The results are expressed as median fluorescence of all the beads. For IgA and IgM studies, each sample is depleted of IgG to ensure specificity.

A set of samples randomly picked were run 10x to control for internal variation when establishing the assay. The Luminex assay has a considerable number of internal controls that the sample needs to pass prior to obtaining valid results. The assay must have a minimum number of antigen and control beads. Total IgG, IgA, and IgM are evaluated to avoid over dilution and the associated issues Reviewer 2 is mentioning. Internal software will flag samples that do not pass these QA/QC and samples will be repeated. All results are reviewed by three laboratory team members including the first and last authors.

Finally, we evaluated 103 cord blood samples collected prior to SARS-CoV-2 to establish thresholds and used a very conservative threshold (see Table S1 regarding the statistical risk of a false positive).

4. Was there any consideration to performing variant-specific SARS-CoV-2 IgA/IgM antibody testing? This would strengthen the study's findings regarding variant-specific risk of in utero transmission.

Response: Variant-specific SARS-CoV-2 antibody testing was not available at the time of this experiment and would be difficult to implement in such a large population. Variants change so rapidly, and we did not perform neutralizing antibody testing for this study.

5. It would be helpful if the authors could comment more on the use of IgA antibodies as an indicator of fetal infection. Are there other infections that use cord blood IgA antibodies to diagnose fetal infection? At what gestational age do fetuses begin making IgA antibodies in response to fetal infection?

Response: Diagnosis for other congenital infections, such as toxoplasmosis and Rubella have used multi-isotype panels that include IgA. In general, TORCH infections are confirmed by serology (see References 25-28). Zika Virus infection is another congenital infection that can be confirmed by serology (covered in reference 25).

Fetal IgA can be detected as early as 24-27 weeks (see references #27 and #30).

6. Although we understand that the cord blood samples are deidentified, were the authors able to collect any clinical information on these patients, such as fetal sex or gestational age at delivery? If so, this would be helpful to include in the manuscript.

Response: A subset of the participants (n=400) consented to this study at birth and data collection is ongoing. We do have demographic and medical history for these participants and plan to present these data in a future manuscript once we have enrolled more participants with adequate follow-up. However, because it is only a subset of the total population, we did not include here as we cannot be sure it is representative of the total population studied.

For the other samples, we have no information.

7. What is the false positive rate of IgA antibodies? IgM antibodies are known to have high false positive rates, and diagnosis of in utero SARS-CoV-2 infection generally requires multiple positive IgM tests collected over time with a pattern that serially declines.

Response: As described in the manuscript, adopting the more conservative threshold (LoB+5×SDblank) reduced false positives to 0.001% for IgA anti-RBD, 0.003% for IgA anti-N, 0.02% for IgM anti-RBD, and 0.006% for IgM anti-N in the simulation results (Table S1). The probability of falsely detecting SARS-CoV-2 antibodies in pre-pandemic samples is exceedingly low, enhancing the specificity of the testing method. The presence of IgM antibody at birth depends on timing of infection. We have preliminary data (unpublished) demonstrating decline of IgM in newborns. However, routine testing for COVID is not done currently, but with screening of all exposed newborns, they may be followed for further testing and follow-up for any negative outcomes.

8. What it be possible to confirm how many patients that were IgA/IgM antibody positive were also PCR positive?

Response: For the vast majority it is impossible, as the samples were de-identified. We have some partial information for participants who consented at birth to the ongoing study mentioned above, but we do not have enough data for a statistically relevant analysis. PCR would only be helpful if positive during pregnancy or at delivery.

9. Lines 223-224: It is not necessarily true that anti-RBD and anti-N IgG antibodies are indicative of maternal infection during pregnancy. Maternal IgG antibodies persist indefinitely and cross the placenta, thus the presence of anti-RBD and anti-N IgG antibodies may be due to a remote infection prior to pregnancy.

Response: This is correct. We have clarified in the text that maternal infection could have occurred any time prior to delivery including prior to pregnancy. While it is true that IgG antibodies to anti-RBD and anti-N are present after infection, there is decline in both antibodies and in fact some people can turn negative over time. We present data showing a statistically significant relationship between IgM and IgA levels with maternal IgG. IgG levels are higher when IgA or IgA and IgM are present, compared to lower IgG levels when only IgM is present. This is supporting evidence of timing of infection.

10. Are the authors able to comment on the impact of maternal vaccination status on the results?

Response: Unfortunately, we cannot yet. This is one of the goals of the ongoing study mentioned above. Right now, we do not have enough data for a statistically relevant analysis. That will be a separate manuscript when more data has been collected.

11. Lines 282-283: Although the presence of SARS-CoV-2 specific IgA antibodies in neonatal cord blood samples is suggestive of in utero infection, we do not feel that the results of this study are sufficient to conclude that in utero infection has occurred. The study is limited by having a single isolated sample (rather than serial infant samples), the potential for false positive IgA/IgM results, the possibility of maternal blood contamination in cord blood samples, and the absence of PCR data to confirm in utero infection. We agree that IgA antibodies should be explored in future work as a potential marker for in utero transmission, but we feel the conclusions should be softened as this study does not provide sufficient evidence of in utero COVID-19 infection.

Response: We believe that it is essential to have a screening tool that can identify newborns with potential IU-SC and the fetal antibody response in cord blood is one of the few tests that can be easily evaluated on all births. This is imperative so that they can then be closely monitored and followed up for repeat testing and evaluation for neurodevelopmental and other abnormalities. Current data are suggesting that exposed newborns may have developmental abnormalities and other abnormalities. Follow-up of infants identified by IgA/IgM and PCR would allow further close follow up with repeat testing.

The conclusions have been edited and softened to reflect Reviewer 2's comment.

Response to Reviewer #2

I feel that the emphasis is still too strong that IgA positivity in cord blood is indicative of in utero transmission. I agree with the prior reviewer that it is indicative of fetal response to maternal infection, as viral proteins or antigens (not infectious virus) may cross the placental barrier and stimulate the fetal immune response.

This has been edited throughout the manuscript. We do not refer to “in utero transmission” and have replaced it by “fetal immune response”.

Testing of a single sample and not in duplicate, the experimental rigor is reduced. The vast majority of Luminex bead-based assays are conducted in duplicate or even triplicate, per the manufacturer's recommendations.

For this specific assay, Luminex does not recommend testing in duplicate/triplicate. To ensure reproducibility, Luminex requires that at least 350 beads (>50 beads for each of the 7 antigens tested) be analyzed, to yield reliable fluorescence results. Each bead yields an individual result. The data presented in the manuscript is the Median Fluorescence Intensity (MFI) of >50 individual beads for each specific antigen. If a sample did not reach these levels, it was flagged by the Luminex xPonent software and repeated.

During our validation process, we internally confirmed that >350 beads yielded reliable results by running replicates of control samples from the WHO, Bio-Rad, and Luminex on every run. We also ran replicates of randomly selected plasma/serum samples.

The specificity of the assay has also been evaluated and confirmed by other groups (Cox *et al.*, 2023 - PMID: 37059296).

Finally, to avoid false positives due to assay variability on samples close to the threshold, we decided on a conservative threshold, using a limit of detection with 5 standard deviations above the limit of blank, rather than the 1.645 fold usually used. This reduced, even further, the risk of false positives (see Supplementary Table).

The response to the reviewer #2 has been included in the manuscript, method section, line 372-374:

Assay validation included running randomly selected samples multiple times and WHO standards within each run to ensure reproducibility.

We also included 2 references from Luminex regarding this specific assay (references #19 and #20).